# Membranes linked by *trans*-SNARE complexes require lipids prone to non-bilayer structure for progression to fusion

Michael Zick[1], Christopher Stroupe[2,3], Amy Orr[1], Deborah Douville[1], William T Wickner[1]*

[1]Department of Biochemistry, Geisel School of Medicine at Dartmouth, Hanover, United States; [2]Department of Molecular Physiology and Biological Physics, University of Virginia, Charlottesville, United States; [3]Center for Membrane Biology, University of Virginia, Charlottesville, United States

**Abstract** Like other intracellular fusion events, the homotypic fusion of yeast vacuoles requires a Rab GTPase, a large Rab effector complex, SNARE proteins which can form a 4-helical bundle, and the SNARE disassembly chaperones Sec17p and Sec18p. In addition to these proteins, specific vacuole lipids are required for efficient fusion in vivo and with the purified organelle. Reconstitution of vacuole fusion with all purified components reveals that high SNARE levels can mask the requirement for a complex mixture of vacuole lipids. At lower, more physiological SNARE levels, neutral lipids with small headgroups that tend to form non-bilayer structures (phosphatidylethanolamine, diacylglycerol, and ergosterol) are essential. Membranes without these three lipids can dock and complete *trans*-SNARE pairing but cannot rearrange their lipids for fusion.

*For correspondence: William.T.Wickner@dartmouth.edu

Competing interests: The authors declare that no competing interests exist.

## Introduction

Membrane fusion on the exocytic and endocytic pathways underlies cell growth, hormone secretion, neurotransmission, and certain nutrient and pathogen uptake (*Wickner and Schekman, 2008*). Fusion is guided and catalyzed by families of proteins which are conserved from yeast to humans: Rab family GTPases serve as membrane-identity master switches (*Grosshans et al., 2006*), binding large multisubunit effector complexes which catalyze tethering, the first step of specific membrane association. Membranes bear SNARE proteins (*Jahn and Scheller, 2006*), with characteristic heptad repeats with a central arginyl (R-SNARE) or glutamyl (Q-SNARE) residue (*Fasshauer et al., 1998*) and membrane anchor domains. SNAREs can bind (snare) each other, either as *cis*-SNARE complexes in which each SNARE is anchored to the same membrane or as *trans*-SNARE complexes in which SNAREs are anchored to two different apposed membranes. When the SNAREs of each membrane have been liberated from *cis*-SNARE complexes by the SNARE complex disassembly system of Sec18p (NSF), Sec17p (α-SNAP), and ATP (*Mayer et al., 1996*), and the membranes have been tightly apposed through tethering, SNAREs can assemble into *trans*-SNARE complexes. *Trans*-SNARE complex assembly is a prerequisite for the bilayer rearrangements of fusion.

Despite increasingly detailed knowledge of these protein catalysts, far less is known of the lipid requirements for fusion. Fusion can be reconstituted with a few recombinant SNARE proteins in liposomes of simple lipid composition (*Weber et al., 1998*; *Fukuda et al., 2000*) such as phosphatidylcholine (PC) plus phosphatidylserine (PS). Nevertheless, genetic screens for in vivo fusion defects and fusion assays of isolated organelles emphasize the importance of a more complex lipid composition under physiological conditions.

We study membrane fusion with the vacuoles (lysosomes) of *Saccharomyces cerevisiae* (*Wickner, 2010*). An initial screen for altered vacuole morphology (*Wada et al., 1992*), the *vam* phenotype, identified

**eLife digest** All cells are enclosed with a membrane that is made of phospholipid molecules, and many of the structures found inside cells—such as the vacuoles in plant and fungal cells—are also enclosed with a phospholipid membrane. To form a membrane, the phospholipid molecules—which have a phosphate head and two fatty acid tails—arrange themselves in two layers, with the fatty acid tails pointing into the membrane, and the phosphate heads pointing outwards. This structure is known as a phospholipid bilayer.

Vacuoles are filled with water that contains various proteins and molecules in solution, and adjust their volume to keep the concentrations of substances in the cell in balance. To do this, the vacuoles fuse with each other. This fusion process requires dramatic spatial rearrangements of the phospholipid molecules.

The SNARE family of proteins plays a key role in membrane fusion. As the two membranes come together, SNARE proteins located on each membrane form a complex known as a *trans*-SNARE complex. This docks the vacuole in place beside another vacuole while the phospholipid molecules in the two membranes rearrange. However, much less is known about the phospholipid molecules that are involved in the fusion process.

Now, Zick et al. have shown that three types of phospholipid molecules must be present for membrane fusion to be completed. These have in common that their phosphate 'headgroups' are small and they do not tend to form bilayers. The vacuoles can dock beside each other if these small headgroup phospholipid molecules are not present, but the bilayer lipids in the vacuole membranes cannot rearrange themselves in the absence of these particular lipids.

The importance of these nonbilayer lipid molecules had not previously been established, as the majority of experiments investigating membrane fusion used concentrations of SNARE proteins that were much higher than those found physiologically. At such high concentrations, fusion can go ahead without the nonbilayer lipid molecules being present.

the vacuole-specific Rab (now termed Ypt7p), the vacuole-specific SNAREs, and subunits of the Rab effector complex (termed HOPS, for homotypic fusion and vacuole protein sorting; *Seals et al., 2000*; *Wurmser et al., 2000*). However, later genomic screens for fragmented vacuole morphology in strains with defined nonessential gene deletions suggested that sterol and phosphoinositides were also required for fusion (*Seeley et al., 2002*). With a quantitative, colorimetric assay of the fusion of purified vacuoles, biochemical studies confirmed vital roles for phosphoinositides (*Mayer et al., 2000*; *Cheever et al., 2001*; *Seeley et al., 2002*; *Fratti et al., 2004*; *Mima and Wickner, 2009*; *Xu and Wickner, 2010*), diacylglycerol (*Jun et al., 2004*), and ergosterol (*Kato and Wickner, 2001*; *Seeley et al., 2002*). It was found that each of these lipids co-localized with the Rab, Rab-effector, and SNAREs in the fusion microdomain of docked vacuoles, and that the localization of these lipids to this microdomain is interdependent with localization of the fusion proteins (*Fratti et al., 2004*).

Exploiting an assay of fusion of proteoliposomes consisting of vacuolar lipids, the purified prenylated Rab Ypt7p, 4 recombinant vacuolar SNAREs (Vam3p, Vti1p, Vam7p, and Nyv1p), HOPS, Sec17p, Sec18p, and ATP (*Zucchi and Zick, 2011*), we have reexamined the roles of lipids in the fusion reaction. We find that small head-group neutral lipids that tend to form nonbilayer structures are essential for fusion at physiological SNARE concentrations. Small head-group neutral lipids are not needed for *trans*-SNARE pairing, but are required for the ensuing lipid rearrangements that constitute membrane fusion.

## Results

Membrane fusion was measured as protected lumenal compartment mixing and concurrent lipid mixing by a modified version of our published assay (*Zucchi and Zick, 2011*). Reconstituted proteoliposomes (RPLs) were prepared with a recombinant prenylated Rab (Ypt7p) and with the four vacuolar SNAREs (Nyv1p, Vam3p, Vti1p, and Vam7p, which are the vacuolar R, Qa, Qb, and Qc SNAREs; *Fasshauer et al., 1998*). RPLs were formed from an octylglucoside mixed-micellar solution of these proteins and vacuolar lipids during lengthy dialysis in the cold, then isolated by flotation. One set of proteoliposomes (*Figure 1A*) bears Marina Blue-linked phosphatidylethanolamine as a lipidic marker and entrapped Cy5-derivatized

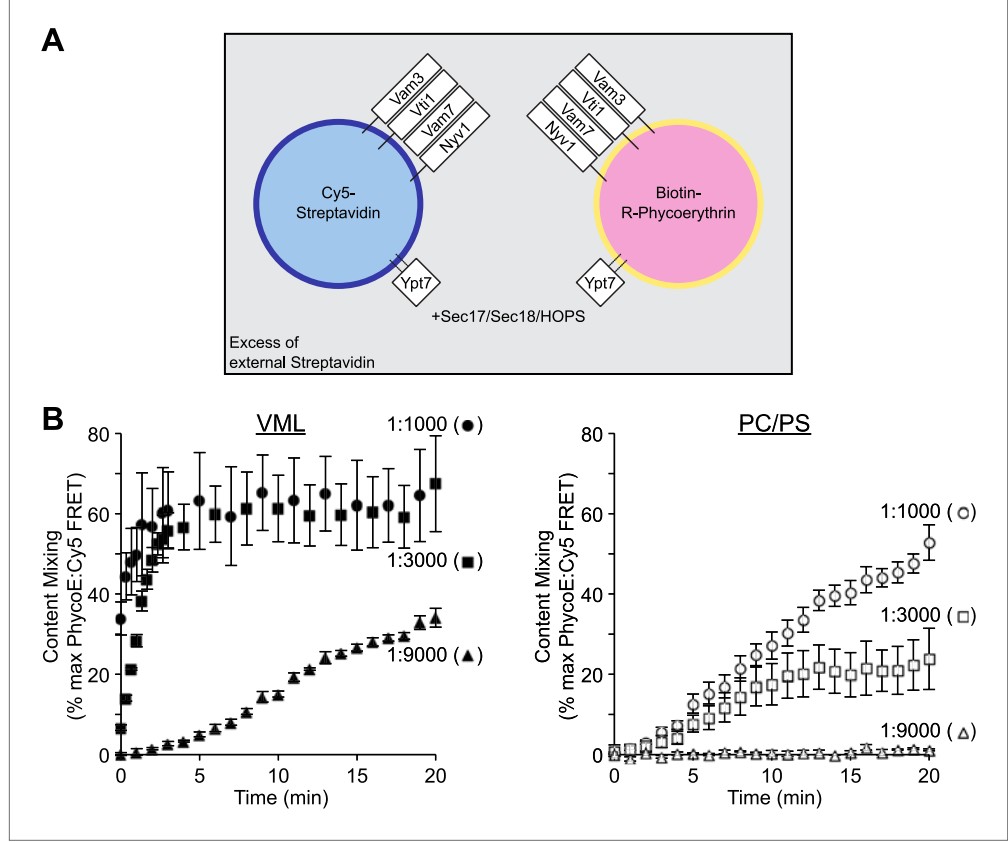

**Figure 1**. Lipid composition and SNARE concentration regulate the rate of proteoliposome membrane fusion. (**A**) Membrane fusion was assayed as protected (from external non-fluorescent streptavidin) lumenal compartment mixing. This was measured as the FRET between biotin-phycoerythrin and Cy5-streptavidin, which had been entrapped within separate proteoliposome populations. Paired sets of proteoliposomes were prepared with either the complete vacuolar mixed lipids (VML) or with 70% PC/30% PS. Proteoliposomes bore Ypt7p and the 4 SNAREs, each at a 1:1000, 1:3000, or 1:9000 molar ratio to lipid phosphate, as described in the 'Materials and methods'. For each pair, half the proteoliposomes had 0.3% of its lipid as Marina Blue-PE and bore lumenal Cy5-streptavidin, while the complementary proteoliposomes had 1.5% NBD-PE and bore lumenal biotinylated-phycoerythrin. (**B**) Fusion assays were performed with Sec17p, Sec18p, HOPS, and ATP in the presence of excess nonfluorescent streptavidin, as described in the 'Materials and methods'. Error bars here and in subsequent figures are the standard deviations from three assays.

streptavidin as a lumenal marker, while a complementary set of RPLs bears the lipidic marker NBD-PE and lumenally entrapped biotinylated phycoerythrin. Fusion reactions are performed in the presence of a large excess of external, nonfluorescent streptavidin to bind any biotinylated R-phycoerythrin that may be released from the proteoliposomes by lysis. Upon addition of purified HOPS, Sec17p, Sec18p, and $Mg^{2+}$:ATP, fusion allows the biotin-R-phycoerythrin to bind to the Cy5-streptavidin within the lumen of fused vesicles while remaining inaccessible to the external, nonfluorescent streptavidin. This is readily assayed by the ensuing Förster resonance energy transfer (FRET) between the Cy5 and R-phycoerythrin. Fusion is also reflected in the quenching of the fluorescence of Marina Blue-PE as it is mixed in the same bilayer with NBD-PE. The rate and extent of fusion are governed by the proteoliposomal lipid composition and by the molar ratio of SNARE proteins to lipid. Proteoliposomes of a vacuolar mixed lipid (VML) composition, based on the established composition of the isolated organelle (*Schneiter et al., 1999*; *Zinser et al., 1991*), bearing SNAREs at a 1:1000 molar ratio to lipids (*Figure 1B*, left, filled circles) or at a 1:3000 ratio (filled squares) undergo rapid fusion. At a 1:9000 SNARE:lipid ratio, the fusion of VML proteoliposomes (filled triangles) slows to the rate seen for PC/PS proteoliposomes at a 1:1000 ratio (*Figure 1B*, right, open circles). Fusion is hardly detectable for PC/PS proteoliposomes with SNAREs at a 1:9000 molar ratio to lipids (open triangles).

To place these findings in a context of the physiological concentrations of SNAREs, vacuoles were purified (*Haas, 1995*) and analyzed for lipid phosphorus and for their bound Ypt7p, HOPS, Sec17p, Sec18p, and each of the 4 SNAREs. These proteins were from 5- to 100-fold less abundant on vacuoles as compared to proteoliposomes which were prepared with a 1:1000 SNARE:lipid molar ratio (*Table 1*) and in which approximately half the SNAREs were shown by protease-accessibility assay (*Figure 2C*) to be exposed on the proteoliposome exterior. Thus the lower end of SNARE concentrations employed in our reconstituted reactions, while still high compared to the organelle, are closer to physiological. Only very high SNARE concentrations can partially bypass the requirement for greater lipid complexity for fusion.

To evaluate the role of lipids in vacuolar fusion, we started from the complete VML composition (*Mima et al., 2008*) with SNAREs at a 1:5000 molar ratio to lipids and sequentially removed one lipid at a time, substituting additional PC in its place. The sequential removal of PE, diacylglycerol, and ergosterol reduced the rate and extent of fusion, until fusion could no longer be detected when all three of these lipids were omitted (*Figure 2A*). These proteoliposomes had comparable lumenal entrapment of fluorescent proteins (data not shown), comparable protein composition (*Figure 2B*), and comparable orientation of SNARE and Rab proteins, as judged by protease accessibility assays (*Figure 2C*). Proteoliposomes vary in size from preparation to preparation, and according to their SNARE composition. Those prepared without the three non-bilayer lipids have approximately 20% smaller diameter, though this is still largely within the range seen for VML RPLs of varying SNARE composition (*Figure 2D*). Thus, the lack of fusion signal was presumably not due to an absence of the Rab, SNAREs or lumenal probe, or to altered proteoliposome topology.

Vacuoles with 3- to 9-fold elevated SNARE levels undergo lysis as well as fusion (*Starai et al., 2007*), and vacuolar proteoliposomes also exhibit both behaviors (*Zucchi and Zick, 2011*). Lysis is inferred from the extra FRET obtained from the initially-lumenal probes when the external quencher, nonfluorescent streptavidin, is omitted (*Zucchi and Zick, 2011*). Proteoliposomes with the full VML lipid composition undergo fusion and lysis (*Figure 3*, squares), whereas RPLs lacking PE, DAG, and ERG exhibit neither fusion nor lysis (open and filled circles). Thus the lack of fusion signal when PE, ERG, and DAG are absent is not due to a fusion pathway diversion into lysis.

The vacuole fusion pathway entails early ATP-dependent reactions that occur on separate vacuoles, termed priming, followed by tethering, which can be mediated by HOPS and Ypt7p alone (*Hickey and Wickner, 2010*). Tethering allows a striking enrichment of fusion proteins and lipids in a microdomain, followed by *trans*-SNARE complex assembly. This is followed by rearrangements of the lipid bilayers

**Table 1.** Protein abundance, relative to lipids, in vacuoles or reconstituted proteoliposomes (RPL) fusion reactions

| Protein | Molar ratio of lipid:protein in RPL reactions* | Molar ratio of lipid:protein on vacuoles | | Ratio (RPLs/vacuoles) of molar protein:lipid ratios in std. reactions† |
|---|---|---|---|---|
| | | BJ3505 | DKY6218 | |
| Vam7p | $2 \times 10^3$ | $30 \times 10^4$ | $6.5 \times 10^4$ | $7 \times 10^1$ |
| Vam3p | $2 \times 10^3$ | $11 \times 10^4$ | $22 \times 10^4$ | $7 \times 10^1$ |
| Vti1p | $2 \times 10^3$ | $10 \times 10^4$ | $13 \times 10^4$ | $5 \times 10^1$ |
| Nyv1p | $2 \times 10^3$ | $4.3 \times 10^4$ | $8.1 \times 10^4$ | $3 \times 10^1$ |
| Ypt7p | $4 \times 10^3$ | $1.9 \times 10^4$ | $1.8 \times 10^4$ | $0.5 \times 10^1$ |
| Sec17p | $7 \times 10^3$ | $41 \times 10^4$ | $13 \times 10^4$ | $3 \times 10^1$ |
| Sec18p | $1 \times 10^3$ | $10 \times 10^4$ | $13 \times 10^4$ | $10 \times 10^1$ |
| Vps33p | $6 \times 10^3$ | $17 \times 10^4$ | $31 \times 10^4$ | $3 \times 10^1$ |

*for SNAREs and Ypt7p, calculated, based on a 1:1000 lipid:protein ratio during reconstitution and an assumption of 50% outwardly-oriented SNAREs on proteoliposomes; for others, based on amounts of added proteins, and 0.74 mM lipids in standard proteoliposome reactions (see 'Materials and methods').

†based on measured (see 'Materials and methods') values of 2.17 nmol lipid per µg total vacuole protein for BJ3505 vacuoles and 1.00 nmol lipid per µg total vacuole protein for DKY6218 vacuoles, and standard vacuole reactions containing 3 µg protein of each vacuole in 30 µl.

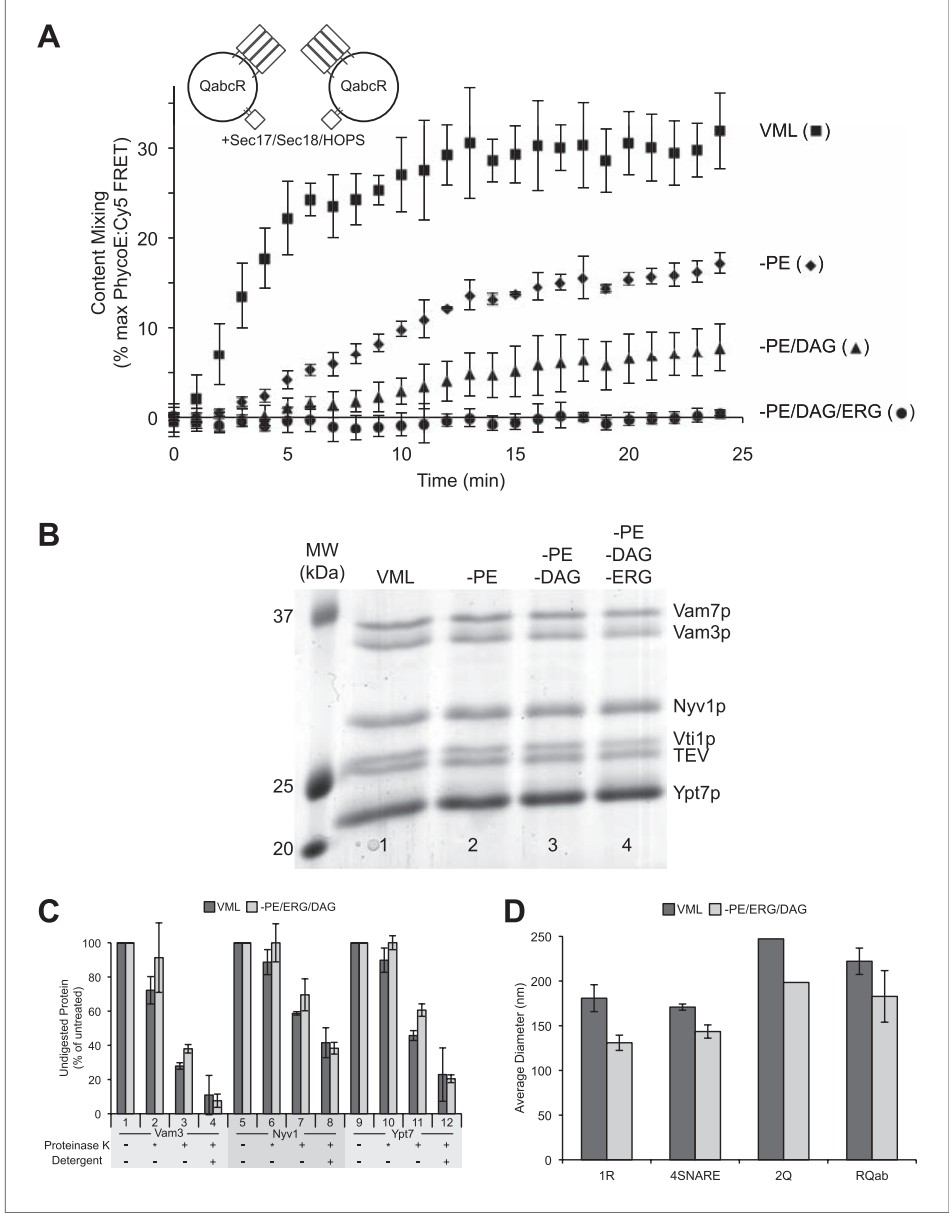

**Figure 2**. A role for neutral, non-bilayer lipids in fusion. (**A**) Fusion of proteoliposomes bearing Ypt7p and 4-SNAREs (1:5000 molar ratio to lipid phosphate), prepared with either the complete vacuole lipid mix (squares) or missing PE (diamonds), PE and DAG (triangles), or PE, DAG, and ERG (circles). Fusion was assayed as the FRET between lumenal fluorescent proteins in the continuous presence of an excess of external nonfluorescent streptavidin. (**B**) Proteoliposomes were analyzed for their protein composition by SDS-PAGE and Coomassie blue staining. Lipid composition: Lane 1, complete vacuolar lipid mix; lane 2, PE omitted; lane 3, PE and DAG omitted; lane 4, PE, DAG, and ERG omitted. In each case, the percentage of PC was increased to account for the omitted lipid(s). (**C**) Similar protease-accessibility of SNAREs and Rab across proteoliposomes preparations. Proteoliposomes (1.2 mM lipid) were incubated in 15 µl RB150 with 60 mM HEPES/NaOH pH 8.0 for 10 min at 27°C with either no addition of protease, with 60 µg/ml of proteinase K which had been preincubated for 10 min with 1 mM PMSF prior to proteoliposome addition (indicated by asterisk), with fully-active proteinase K which had not been preincubated with PMSF, or with fully active proteinase K and 1% (wt/vol) β-octylglucoside. After this incubation, PMSF was added to samples which had fully active proteinase K and the incubation continued for an additional 10 min. All samples were then mixed with SDS sample buffer, heated to 95°C for 5 min and subjected to SDS-PAGE. Gels were stained with Coomassie blue, and bands corresponding to Vam3p, Nyv1p, and Ypt7p quantified by scanning with a Microtek Bio-5000 scanner (Microtek Lab, Inc., Santa Fe Springs, CA) and UN-SCAN-IT gel 5.3 software (Silk Scientific,

*Figure 2. Continued on next page*

*Figure 2. Continued*

Orem, UT). For each of these three proteins, the intensity of the band from samples which never saw proteinase K was set to 100%. Dark bars correspond to VML proteoliposomes, light bars to proteoliposomes prepared without PE, DAG, and Erg. Shown is the average of two experiments +/− standard deviations. (**D**) The size distribution of various proteoliposome preparations was analyzed by dynamic light scattering with a Zetasizer nano ZS (Malvern Instruments Inc., Westborough, MA) through non-invasive back-scatter at 173°. For each liposome preparation, at least four samples (400 µl at a lipid concentration of 20 µM) were measured in low volume disposable sizing cuvettes at 25°C. Shown is the average diameter (+/− standard deviation) of independent proteoliposome preparations composed of the complete vacuolar lipid mix (dark bars) or without PE, ERG, DAG (light bars) and bearing either Nyv1 (1R; n = 3), all four SNAREs (4SNARE; n = 2), Vam3 and Vti1 (2Q; n = 1), or Nyv1, Vam3, and Vti1 (RQab; n = 2).

resulting in membrane fusion and the attendant mixing of lumenal compartments. To determine the stage(s) that require the nonbilayer-prone-lipids, paired sets of proteoliposomes were prepared with VML lipids, Ypt7p, and either the R-SNARE Nyv1p or the Q-SNAREs Vam3p (Qa) and Vti1p (Qb). These proteoliposomes readily fuse when incubated with HOPS and Vam7p (Qc-SNARE); Sec17p and Sec18p stimulate but are not required (*Zick and Wickner, 2013*) and were not present in these assays. A second set of proteoliposomes was prepared in parallel, but lacking PE, DAG, and ERG. There was no fusion in the absence of the three small headgroup lipids (*Figure 4A*). After a 10 min incubation under fusion conditions, aliquots were solubilized in a RIPA buffer (1% Triton X-100, 1% sodium cholate, and 0.1% SDS) with affinity-purified antibody to Vam3p, then mixed with magnetic beads bearing protein A. After washing, bound proteins were eluted with hot SDS and analyzed for Nyv1p by immunoblot (*Figure 4B*). Comparable amounts of Nyv1p had bound to Vam3p in proteoliposomes with complete vacuolar lipid mix (*Figure 4B*, lane 2), where rapid fusion occurred, as in proteoliposomes lacking PE, DAG, and ERG (lane 5) where there was no detectable fusion (*Figure 4A*). Nyv1p did not associate with Vam3p in *trans* when Vam7p was omitted (*Figure 4B*, lanes 1,4) or when Vam7p was only added immediately after the RIPA buffer (lanes 3, 6).

PE, DAG, and ERG are required for fusion at SNARE:lipid molar ratios of 1:5000, whether the proteoliposomes bear all 4 SNAREs (*Figure 2A*) or 2Q-RPLs and 1R-RPLs are incubated with added

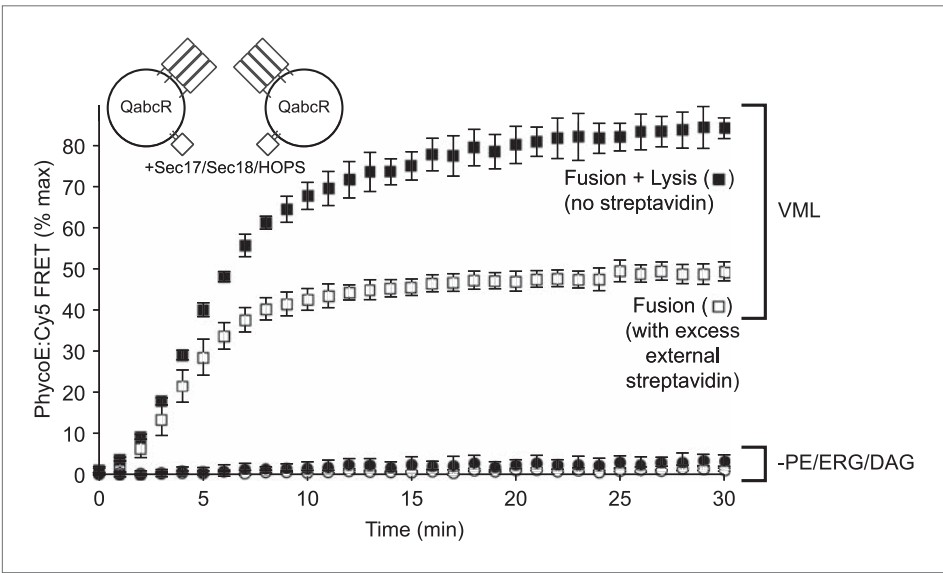

**Figure 3**. Omission of PE, DAG, and ERG blocks HOPS, Sec17p, and Sec18p triggered lysis as well as fusion. Proteoliposomes of complete vacuolar lipid mix or lacking PE, DAG, and ERG and with Ypt7p and the 4 vacuolar SNAREs (1:5000 molar ratio to lipid phosphate) were incubated either with a large molar excess of non-fluorescent streptavidin, restricting FRET development to sealed fusion events, or without external streptavidin, yielding FRET from both fusion and from lysis (*Zucchi and Zick, 2011*).

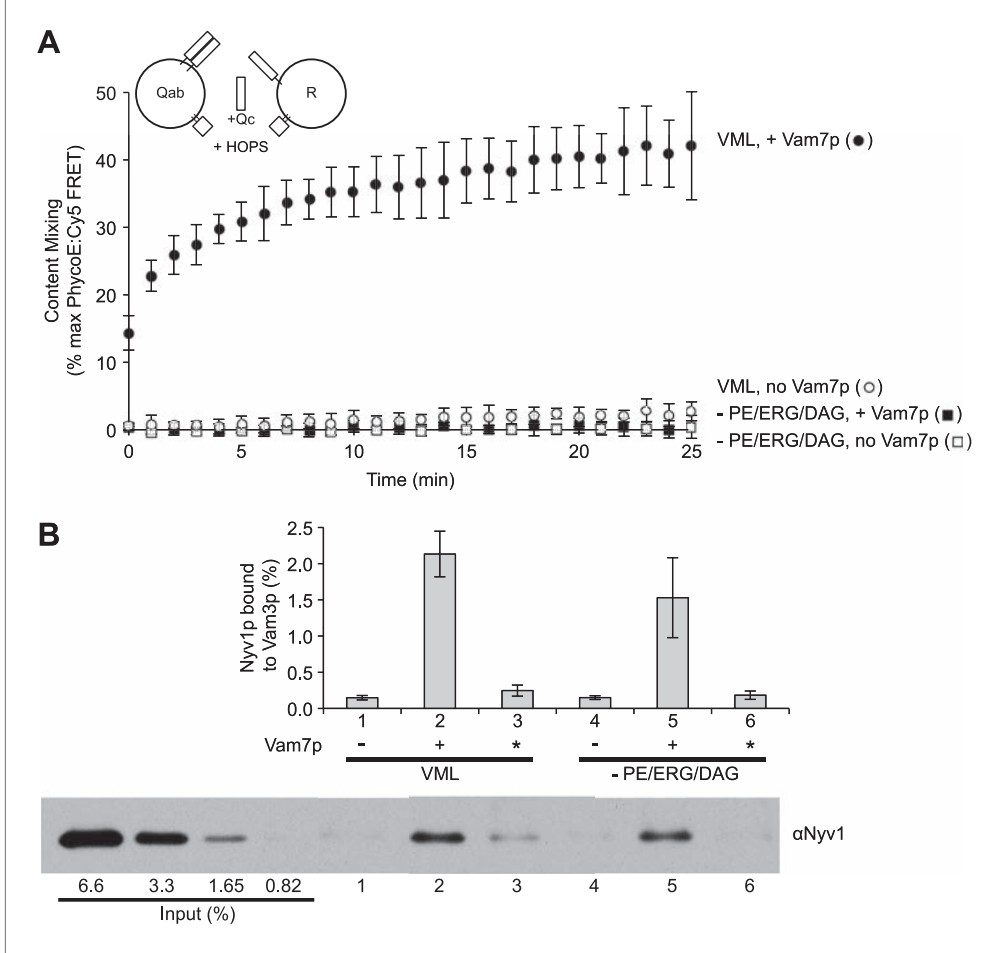

**Figure 4**. Small-headgroup, nonbilayer lipids are needed for trans-SNARE docked membranes to proceed to fusion. Reconstituted proteoliposomes with either the R-SNARE or the Vam3p and Vti1p Q-SNAREs, prepared at a 1:5000 molar ratio of SNARE to lipid and either having the complete vacuolar lipid mix or without PE, ERG, or DAG were incubated in fusion reactions. Vam7p (0.5 μM) was added where indicated, either during the fusion reaction (lanes 2 and 5) or after the reaction was terminated by detergent addition (indicated by an asterisk, lanes 3 and 6). Each reaction was (**A**) assayed for lumenal content mixing and (**B**) mixed after 10 min with a 10-fold volume of a modified RIPA buffer (20 mM HEPES/NaOH, pH 7.4, 0.15M NaCl, 0.2% bovine serum albumin (defatted), 1% Triton X-100, 1% sodium cholate, 0.1% sodium dodecyl sulfate, 1 mM EDTA) with 40 μg/ml affinity-purified antibody to Vam3p and 1 μM recombinant soluble domain of Snc2p to suppress SNARE complex assembly in detergent. After addition of 10 μl of RIPA buffer-washed suspension of magnetic beads with bound protein A (Thermo Scientific), samples were mixed for 1 hr at room temperature. Beads were collected by placing the tubes for 2 min onto a magnetic rack, and the unbound proteins removed. Beads were thrice washed with 1 ml of modified RIPA buffer, then proteins were eluted with SDS sample buffer at 95°C and analyzed by SDS-PAGE and immunoblot with antibodies to Nyv1p. Reactions were performed without further SNARE addition, with 0.5 μM Vam7p from the start of the incubation, or with the Vam7p added one minute after solubilization by RIPA buffer. The same preparations and solutions were premixed, then used in parallel for the assays of fusion and *trans*-assembly of SNAREs shown here. The immunoblot of one of the three *trans*-SNARE assays is shown.

Vam7p (**Figure 4A**). It remained possible that some other parameter of reconstitution which we could not measure, such as the *trans*-membrane disposition of each lipid species, might regulate fusion and be influenced by the SNARE levels. To address this possibility, we prepared RPLs bearing both the R-SNARE and QaQb (Vam3p, Vti1p) SNAREs, with either the complete mixture of vacuolar lipids or lacking PE, DAG, and ERG. In either case, fusion of these RQaQb RPLs was not seen without added Vam7p (**Figure 5**, diamonds). With ample added Vam7p (5 μM) to allow maximal formation of *trans*-SNARE complexes, the fusion only showed a several-fold stimulation by the presence of PE, DAG, and

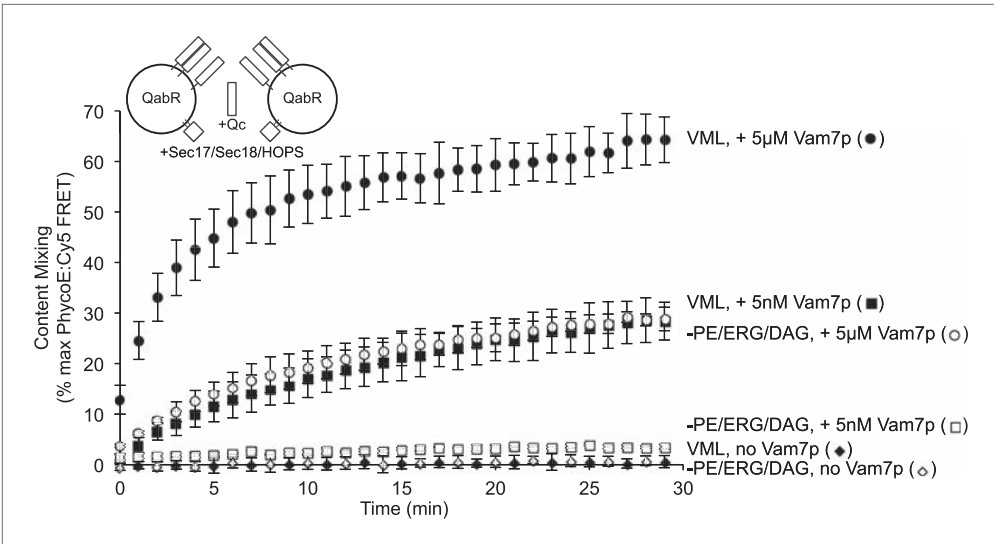

**Figure 5**. The requirement for PE, ERG, and DAG is governed by the level of trans-SNARE complex. Fusion assay pairs of proteoliposomes were prepared with Ypt7p, with VML lipids or lacking PE, ERG, and DAG, as indicated, and with Nyv1p, Vam3p, and Vti1p (the R- and Qa- and Qb-SNAREs, respectively) at a 1:1000 molar ratio to total lipid. Fusion assays were initiated by the addition of HOPS, Sec17p, and Sec18p, as described in the 'Materials and methods', as well as the indicated (final) concentration of Vam7p.

ERG (circles). However, with 5 nM added Vam7p, the small headgroup neutral lipids were still required (squares). Thus the very same proteoliposomes could be used to demonstrate a strict requirement for small-headgroup lipids when the *trans*-SNARE complex levels were limited by the 5 nM Vam7p, while with ample Vam7p (5 µM) and hence high *trans*-SNARE levels, this requirement was partially bypassed. Bypass thus directly reflects the formation of high levels of *trans*-SNARE pairs.

## Discussion

The vacuole fusion pathway has well-defined, sequential steps. After ATP-dependent priming, vacuoles undergo tethering. The Rab Ypt7p directly binds the oligomeric tethering complex HOPS (***Seals et al., 2000***). This interaction is necessary (***Stroupe et al., 2006***) for tethering of vacuolar membranes, and is sufficient for tethering of proteoliposomes of a simple lipid composition such as PC/PS (***Hickey and Wickner, 2010***). It was therefore not surprising that comparable levels of trans-SNARE complex formed in the presence or absence of small headgroup lipids (***Figure 4B***), but the need for these particular lipids for the ensuing fusion was unexpected. It has been suggested that fusion occurs immediately and inexorably upon *trans*-SNARE complex formation. However, several studies have suggested that fusion is more complex. Brunger and colleagues. (***Kyoung et al., 2011***; ***Diao et al., 2012***) have shown long kinetic separation between formation of some trans-SNARE complexes and complete membrane fusion for certain spontaneous fusion events (for example, in the absence of calcium) and in the presence of both synaptotagmin and complexin; upon calcium triggering, a burst of fast fusion events were then observed. During yeast vacuole fusion, lumenal compartment mixing occurs many minutes after docking is complete (***Mayer et al., 1996***). Furthermore, when a single SNARE (Vam7p) is withheld from tethered vacuole clusters, the addition of this SNARE triggers rapid and synchronized completion of docking, as assayed by the acquisition of resistance to added antibody to Ypt7p or to Vam3p, but lumenal compartment mixing only occurs far more slowly (***Merz and Wickner, 2004***). Lipid ligands have been reported to block fusion after docking (***Collins and Wickner, 2007***). We now exploit the fully reconstituted vacuole fusion reaction to show that small-headgroup lipids that are prone to non-bilayer structures play a critical role in the progression of docked, *trans*-SNARE paired proteoliposomes to fusion.

The essence of membrane fusion is the rearrangement of lipids from closely apposed bilayers. Models of this rearrangement, such as through hemifusion, posit intermediate states with locally crowded lipid headgroups. This headgroup crowding may constitute a substantial portion of the activation energy of the fusion process (***Chernomordik et al., 1998***; ***Gaudin, 2000***). Lipids such as PE,

ergosterol, and diacylglycerol would fit into these otherwise crowded regions far better than a cylindrical lipid such as phosphatidylcholine.

Upon vacuole docking, the proteins and lipids that are required for fusion become highly enriched at a ring-shaped microdomain surrounding the apposed regions of the docked membranes (*Wang et al., 2002, 2003*; *Fratti et al., 2004*). Although the surface concentrations of proteins required for proteoliposome fusion are higher than those on the vacuole (*Table 1*), they may be comparable to those in this ring-shaped, fusion-competent vacuole microdomain. Additional studies will be required to precisely determine the physiological concentrations of these proteins and lipids in this microdomain.

Previous studies of fusion, whether of vacuoles or of other model systems, have indicated a role for the three vacuolar, non-bilayer-prone lipids. Though the effects of disrupting PE biosynthesis in yeast are likely too pleiotropic to allow examination of its effects on vacuole fusion in vivo, PE is required for the fusion of mitotic Golgi membranes (*Pécheur et al., 2002*). In model studies with recombinant neuronal SNAREs, PE increased the probability of docking but reduced fusion (*Domanska et al., 2010*). These studies differed from those reported here in several crucial aspects: (i) The use of planar lipid bilayers vs highly curved proteoliposomes, (ii) neuronal vs vacuolar SNAREs, (iii) the absence of a Rab and Rab-effector docking system or of an SM protein in the neuronal reconstitution, and (iv) the composition and complexity of lipids in these reconstitutions. Yeast mutants defective in sterol biosynthesis have multiple small vacuoles, indicative of deficient organelle fusion (*Kato and Wickner, 2001*; *Seeley et al., 2002*). The fusion of vacuoles purified from wild-type yeast is blocked by extraction of ergosterol by β-methylcyclodestrin, and fusion can be restored by incubation with cholesterol-loaded β-methylcyclodestrin (*Kato and Wickner, 2001*). Cholesterol has also been shown to be required for Semliki Forest virus fusion (*White and Helenius, 1980*). Diacylglycerol is also directly implicated in vacuole fusion. Whereas a wild-type yeast strain has one or a few vacuoles, deletion of *PLC1*, encoding a phospholipase C which generates diacylglycerol, causes striking vacuole fragmentation (*Jun et al., 2004*), a hallmark of defective fusion (*Wada et al., 1992*). The cell-free fusion of purified vacuoles is blocked by recombinant C1b domain, a DAG ligand, and is stimulated by added Plc1p (*Jun et al., 2004*). Both ergosterol and diacylglycerol become enriched at the fusion microdomain of docked vacuoles, and this enrichment is sensitive to fusion-protein ligands such as Sec17p and an Fab fragment of affinity-purified antibody to Vam3p (*Fratti et al., 2004*). Our current studies with proteoliposomes bearing pure vacuolar proteins and lipids is thus well-grounded in in vivo genetics and morphology as well as cell-free studies of the fusion of the purified organelle.

It is revealing that high concentrations of SNAREs can at least partially bypass the need for non-bilayer-prone lipids. It is currently unclear whether the sole function of *trans*-SNARE pairs is a close and stable tethering of apposed lipid bilayers, or whether they exert a deforming force on each bilayer to provide the energy for lipid spatial rearrangement into non-bilayer intermediates. Prenyl-anchored SNAREs can support the fusion of vacuoles (*Jun et al., 2007*) and of chemically-defined proteoliposomes (*Xu et al., 2011*), and this has recently been extended to synaptic fusion in mice (*Zhou et al., 2013*). These findings suggest that fusion does not depend on force exerted through the formation of continuous α-helices, extending from the SNARE bundle across two *trans*-membrane anchor domains (*McNew et al., 1999*; *Li et al., 2007*). How might high levels of *trans*-SNARE pairs be able to bypass the need for the three small headgroup lipids? A single *trans*-SNARE pair may bring flat or spherical membrane bilayers into close apposition without distorting their bilayer structure, while multiple *trans*-SNARE pairs may draw membranes together over a wider region, creating a bend in the bilayer at the edge of the zone of apposition, as seen for docked vacuoles (*Wang et al., 2002*; *Fratti et al., 2004*). Such a bend may substitute for nonbilayer lipids in initiating the lipid rearrangements of fusion.

Our current working model of fusion is that even single *trans*-SNARE pairs bring membranes into close apposition. The lipid rearrangements for fusion can be driven by the membrane bend induced by multiple *trans*-SNARE pairs, or can occur by the accretion of fusogenic, noncylindrical lipids when the SNARE concentrations are at low, physiological levels. The Vam7p SNARE N-domain has a Phox-homology motif, and has been shown to associate with the *trans*-apposed bilayer (*Xu and Wickner, 2010*) by its affinity for PI(3)P and acidic lipids (*Lee et al., 2006*; *Karunakaran and Wickner, 2013*) and to bear apolar residues which may insert into the bilayer (*Lee et al., 2006*). Similarly, neuronal SNARE-associated synaptotagmin is triggered by $Ca^{2+}$ binding to insert into the lipid bilayer (*Chapman and Davis, 1998*; *Zhang et al., 1998*) and may function to bridge bilayers (*Araç et al., 2006*; *Xue et al., 2008*). These may also cause bilayer distortions that facilitate fusion; there has not been a direct test of whether nonbilayer-forming lipids contribute to the extraordinary rates of calcium-triggered neuronal

fusion. Further tests of this model may rely on the development of assays to measure the physical strain within bilayers, and of reconstituted reactions fusing large proteoliposomes where lipid- and protein-enriched microdomains can be visualized as readily as with vacuoles.

## Materials and methods

### Proteins and reagents

Lipids were obtained from Avanti Polar Lipids, except ergosterol was from Sigma–Aldrich (St. Louis, MO), PI(3)P was from Echelon Biosciences (Salt Lake City, UT), and the fluorescent lipids were from Life Technologies (Carlsbad, CA). Sec18p (*Haas and Wickner, 1996*), Sec17p (*Schwartz and Merz, 2009*), Ypt7p (*Zick and Wickner, 2013*), HOPS (*Zick and Wickner, 2013*), and vacuolar SNARE proteins (*Mima et al., 2008*; *Schwartz and Merz, 2009*; *Zucchi and Zick, 2011*) were purified as described. Vti1p and Nyv1p were exchanged into octylglucoside buffer as described (*Zucchi and Zick, 2011*).

### Vacuole lipid extraction and measurement of vacuole lipid and protein levels

Vacuolar lipids were extracted by a modification of the Bligh-Dyer method (*Bligh and Dyer, 1959*). Chloroform (100 µl) and methanol supplemented with 0.1 M HCl (200 µl) were added to 37 µg vacuoles, as measured by protein content (*Haas, 1995*), in 80 µl RB150+Mg (20 mM HEPES, pH 7.4, 150 mM NaCl, 10% glycerol, 1 mM MgCl$_2$). This single-phase mixture was vortexed thoroughly and incubated at room temperature for 1 hr. RB150+Mg and chloroform (100 µl ea.) were then added. The sample was vortexed thoroughly and centrifuged at 14000×$g$ rpm in an Eppendorf (Hamburg, Germany) 5415C microcentrifuge at room temperature for 30 s. The organic layer was transferred to a 13 × 100 mm round-bottom glass tube (99445-13; Corning Inc., Corning, NY). Chloroform (200 µl) was added to the remaining aqueous layer. This sample was vortexed and centrifuged as above, and the organic layer was removed and added to the organic layer from the first extraction. RB150+Mg (360 µl) and methanol-HCl (400 ml) were added to the combined organic layers. This mixture was vortexed, centrifuged in a Sorvall SpeedVac SC100 (Thermo Fisher Scientific, Waltham, MA) at atmospheric pressure and room temperature for 30 s, and the aqueous layer was removed and discarded.

Vacuole lipid levels were measured using a lipid phosphorus assay. Ammonium molybdate (10 µl of a 2% wt/vol solution) was added to extracted vacuolar lipids, and to standards (0, 5, 10 25, 50, 75, 100, and 125 µl of a 1 mM NaH$_2$PO$_4$ solution). Samples were incubated at 100°C until dry (approximately 1 hr). Perchloric acid (300 µl of a 70% vol/vol solution) was added. Samples were incubated at 180–200°C for 1 hr with occasional vortexing, then cooled to room temperature. Ammonium molybdate (1.5 ml of a 0.4% wt/vol solution) and ascorbic acid (225 µl of a 10% wt/vol solution) were added. Samples were vortexed thoroughly and incubated at 100°C for 10 min, then cooled to room temperature. Absorbance at 820 nm was measured and phospholipid concentration was estimated by comparison of vacuole lipid samples to the phosphate standards. To obtain lipid concentrations for *Table 1*, measured phospholipid concentrations were multiplied by 1.18, to correct for a reported ergosterol:phospholipid molar ratio of 0.18 in vacuolar lipids (*Zinser et al., 1991*).

For estimation of vacuolar protein levels, 6.5 nmol ea. BJ3505 and DKY6218 vacuoles (*Haas, 1995*), here as measured by lipid content (see previous paragraph) rather than protein content, were analyzed by SDS-PAGE and immunoblotting for Vam7p, Vam3p, Vti1p, Nyv1p, Ypt7p, Sec17p, Sec18p, and Vps33p. Protein levels were estimated by comparison of band intensities (measured using a ChemiDoc-It system with LabWorks version 4.5.00.0 software, UVP, Upland, CA) from vacuolar samples to band intensities from standards (3.25, 1.3, 0.65, 0.26, 0.13, 0.052, and 0.026 pmol ea.) of purified recombinant Vam7p, Vti1p, Nyv1p (*Mima et al., 2008*), his$_6$-tagged Vam3 cytosolic domain (*Nichols et al., 1997*), Ypt7p (*Hickey et al., 2009*), his$_6$-tagged Sec17p and his$_6$-tagged Sec18p (*Haas and Wickner, 1996*), and HOPS complex (*Stroupe et al., 2009*).

### Preparation of proteoliposomes

Reconstituted proteoliposomes were prepared as described (*Zick and Wickner, 2013*), with modifications. Chloroform solutions of lipids (vacuolar mixed lipids; VML) were mixed in a glass vial: 49.6 or 51 mol % diC18:2 PC, 15% diC18:2 PE, 1% diacylglycerol, 8% ergosterol, 2% diC18:2 PA, 18% soy PI, 4.4% diC18:2 PS, 1% diC16 PI(3)P and either 0.23% Marina Blue-PE or 1.5% NBD-PE (Life Technologies). When small headgroup lipids were omitted, the amount of PC was adjusted to bring the sum to 100%. β-octylglucoside was added to 160 mM from a 0.5 M solution in methanol and samples were dried

under a stream of nitrogen, then in vacuo. Samples were dissolved in a fivefold concentrate of RB150+Mg (0.1 M HEPES/NaOH, pH 7.4, 0.75 M NaCl, 50% glycerol, 5 mM $MgCl_2$) by several cycles of vortexing for 10 s, rocker mixing for 30 min, and bath sonication for 5 min, yielding mixed micellar solutions with 4 mM lipids and 50 mM detergent. Lipid micellar solutions (200 µl) were mixed with a mixed micellar solution of purified Ypt7p and the indicated SNAREs (550 µl) and 250 µl of either Cy5-derivatized streptavidin (from KPL, Gaithersburg, MD; 8 µM final) or biotinylated phycoerythrin (Life Technologies; 4 µM final). Each ml of solution was added to a rinsed and knotted 6 cm segment of SpectraPor dialysis membrane, 25 kDa cutoff, 7.5 mm diameter (Spectrum Labs, Rancho Dominguez, CA) which was then knotted and dialyzed at 4°C in 250 ml of RB150+Mg (20 mM HEPES, pH 7.4, 150 mM NaCl, 10% glycerol, 1 mM $MgCl_2$ [*Mima et al., 2008*; *Zucchi and Zick, 2011*]) with 1 g of BioBeads SM-2 (Biorad, Hercules, CA) for at least 20 hr with continuous stirring. The isolation of proteoliposomes by flotation was as described (*Zick and Wickner, 2013*). After total phosphate was assayed, samples were brought to 2 mM lipid with RB150+Mg and small aliquots were frozen in liquid nitrogen and stored at −80°C.

### Fusion assays

Assays of proteoliposome fusion and lysis were as described (*Zick and Wickner, 2013*). Adjacent wells of 384-well plates received either the mixed proteoliposomes in RB150+Mg with streptavidin or a mixture of the remaining assay components, which were added to the wells with proteoliposomes after a 10 min preincubation at 27°C. For lysis assays, duplicate wells either received streptavidin or RB150 in its place; the difference between the readings from these wells is a measure of lysis. The final assay component concentrations are: 19.5 mM HEPES/NaOH, pH 7.4, 142 mM NaCl, 11 mM KCl, 1.1 mM imidazole, 9.8% glycerol, 3.3 mM sorbitol, 1.44 mM $MgCl_2$, 0.13 mM 2-mercaptoethanol, 1.1 mM potassium phosphate, 0.17 mM glutathione, 10.7 µM streptavidin, 1.36 mM $Na_2ATP$, 1.84% bovine serum albumin (defatted), 0.000066% Triton X-100, 0.37 mM lipid from each of the 2 proteoliposome populations, 108 nM Sec17p, 0.55 µM Sec18p, and 0.12 µM HOPS.

## Additional information

### Funding

| Funder | Grant reference number | Author |
|---|---|---|
| National Institutes of Health | R01 GM23377-38 | William T Wickner |
| Deutsche Forschungsgemeinschaft | ZI 1339/1-1 | Michael Zick |

The funders had no role in study design, data collection and interpretation, or the decision to submit the work for publication.

### Author contributions

MZ, Acquisition of data, Analysis and interpretation of data, Drafting or revising the article; CS, Conception and design, Acquisition of data, Analysis and interpretation of data; AO, Acquisition of data, Analysis and interpretation of data; DD, Conception and design, Acquisition of data, Drafting or revising the article, Contributed unpublished essential data or reagents; WTW, Conception and design, Acquisition of data, Analysis and interpretation of data, Drafting or revising the article

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
