## [Decision Letter]

Thank you for sending your work entitled “Membranes linked by *trans*-SNARE complexes require lipids prone to non-bilayer structure for progression to fusion” for consideration at *eLife*. Your article has been favorably evaluated by a Senior editor and 3 reviewers, one of whom is a member of our Board of Reviewing Editors.

The following individuals responsible for the peer review of your submission have agreed to reveal their identity: Axel Brunger and Jose Rizo.

The Reviewing editor and the other reviewers discussed their comments before we reached this decision, and the Reviewing editor has assembled the following comments to help you prepare a revised submission.

In this work the effect of lipid composition on vacuolar fusion at various SNARE concentrations is studied. It is found that at physiological concentration (for vacuolar SNAREs), there is a very large effect in the presence of small neutral lipids that are thought to disrupt the lipid bilayer (such as PE, DAG, ERG). Upon higher SNARE concentration, the effect is partially compensated. The starting point for these experiments is the vacuolar lipid mix, and then the individual lipid components are varied. Most importantly, a true content mixing assay was used.

This work shows the critical importance of the nature of particular types of lipids for membrane fusion. Much of the literature on SNARE-dependent membrane fusion has focused on understanding the roles of proteins in inducing and/or regulating fusion, but much less attention has been paid to the lipids and it is clear from studies of other systems that some lipids can be critical for the mechanism of fusion because of their geometry. The authors now have taken advantage of their very extensive characterization of yeast vacuolar fusion and show that fusion requires lipids with small head groups (which favor stalk formation) when using low protein-to-lipid ratios (on the order of 1:5000 or 1:9000). Actually the term low here refers to comparison to what has been commonly used in the literature for reconstitutions, but the authors report in this paper that these protein densities are still comparable or higher than the physiological densities present in vacuoles. The fact that higher SNAREs densities can bypass at least partially the need for the lipids with small head groups is in itself an important finding. It could also suggest different fusion mechanisms or pathways under the different conditions.

Issues to address:

1) The Introduction stresses the requirement in vivo for PI3P in addition to DAG and ergosterol, while the experiments concern the role of PE, DAG and ergosterol in liposome fusion. The role of PI3P is not tested, although the authors allude to a role in recruitment of Vam7 protein. This needs to be clarified since it seems like the authors are switching the lipids under consideration between the Introduction and the Results. Is there a need for PE in vivo? Is the role of PI3P only in Vam7 recruitment? It would be desirable to study a mutant Vam7 protein specifically deficient in PI3P binding in order to address this question.

2) While the authors show that the liposomes made with or without PE, DAG and ergosterol have similar levels of incorporated SNARE proteins, the authors should provide more evidence that these two populations of liposomes are really comparable. Are both populations similar in size? Are the same fractions of SNARE proteins facing out into the media rather than into the lumen?

3) In Figure 4, a comparison is made for reactions with and without PE, DAG and ergosterol with respect to SNARE complex assembly. What about HOPs and Ypt7p recruitment? What about liposome aggregation, which the authors have shown to be a precursor of fusion? A more complete analysis of these two reactions would be desirable.

4) The effect of lipid composition has been studied previously with a simple lipid mixing system and neuronal SNAREs (Domanska, Kiessling, Tamm, Biophys J. 99, 2936, 2010). However, these experiments only considered the effect on docking and lipid mixing at a fixed SNARE concentration. Interestingly, in these experiments, PE had a reducing role of docking/lipid mixing. Clearly, the use of a lipid-mixing assay may have masked the effects on true fusion, so that could be one explanation of the difference. A discussion of the Domanska reference would be appropriate.

5) The results apply to the vacuolar fusion system – for neuronal fusion, the observed SNARE-protein concentration in synaptic vesicles and in active zones is much higher. Nevertheless, the implications of a critical role of PE, DAG, ERG could also apply to the neuronal system since in that case much faster fusion (msec as opposed to min) is required and it is precisely Ca2+ regulated. Some speculations along these lines might be useful.

6) Although the lipid compositions of vacuoles may have already been well documented elsewhere, it would be helpful to the reader to specifically describe the lipid content of yeast vacuoles so that it can be compared with the standard lipid mixture used in the reconstitutions since the composition of the proteoliposomes is the central theme of this paper. If this standard lipid mixture is an accurate reflection of that found in vacuoles, it would be informative to explicitly make that statement.

7) The first sentence of the Discussion suggests the possibility that trans SNARE complex formation induces fusion inexorably. While this view is widely held by many people in the field, I think it would worth mentioning alternate views, in particular because work in Axel Brunger's lab has provided very strong evidence for formation of (partial) trans neuronal SNARE complexes between vesicles without fusion (vesicles can be docked through the SNAREs for 30 minutes without complete fusion!).

8) In the Results section, the text describes a protein-to-lipid ratio of 1:2000, citing Table 1, but Table 1 describes 1:1000. I realize that this apparent discrepancy comes from the fact that only about half of the SNAREs are oriented toward the outside of the vesicles, but readers may still be confused about what the 1:2000 ratio described in the text really means.

---

## [Author Response]

*1) The Introduction stresses the requirement* in vivo *for PI3P in addition to DAG and ergosterol, while the experiments concern the role of PE, DAG and ergosterol in liposome fusion. The role of PI3P is not tested, although the authors allude to a role in recruitment of Vam7 protein. This needs to be clarified since it seems like the authors are switching the lipids under consideration between the Introduction and the Results. Is there a need for PE* in vivo*? Is the role of PI3P only in Vam7 recruitment? It would be desirable to study a mutant Vam7 protein specifically deficient in PI3P binding in order to address this question*.

Since PI(3)P is not a small headgroup lipid, it is not the topic of this paper. Every RPL in this study has PI(3)P; as noted in the Materials & Methods, certain RPLs had small-headgroup lipids omitted, and the amount of PC adjusted accordingly, but PI3P remained constant. We had given too much emphasis to PI3P in the Introduction, causing this confusion; therefore, we've now de-emphasized the recognition between the Vam7p PX domain and PI(3)P in the Introduction, as we and others have published extensively about this.

*2) While the authors show that the liposomes made with or without PE, DAG and ergosterol have similar levels of incorporated SNARE proteins, the authors should provide more evidence that these two populations of liposomes are really comparable. Are both populations similar in size? Are the same fractions of SNARE proteins facing out into the media rather than into the lumen*?

We have addressed this with 2 new pieces of data, dynamic light scattering (which shows that the presence or absence of small-headgroup lipids only causes a small change in proteoliposome size) and assay of proteinase K inaccessibility (which shows that the similar proportions of Vam3p, Nyv1p, and Ypt7p are accessible to proteinase K in the proteoliposomes whether or not they have the 3 small headgroup lipids).

*3) In*
Figure 4*, a comparison is made for reactions with and without PE, DAG and ergosterol with respect to SNARE complex assembly. What about HOPs and Ypt7p recruitment? What about liposome aggregation, which the authors have shown to be a precursor of fusion? A more complete analysis of these two reactions would be desirable*.

Ypt7p, which is prenylated, is incorporated throughout this paper into every RPL as an integrally bound protein during RPL preparation, rather than being recruited. The presence or absence of small headgroup lipids is shown directly to have no effect on Ypt7 incorporation (Figure 2). As we've shown previously, and now cite again in the presentation of Figure 4 in the Results for added clarity, Ypt7p alone binds HOPS with high affinity (35; 17). Ypt7p is sufficient on RPLs of any lipid composition, even PC/PS, to allow vigorous clustering by HOPS (17). The finding (Figure 4) that the RPLs give comparable trans-SNARE pairing is of course in full accord with them having undergone comparable tethering. Proteoliposomes that are trans-SNARE paired have gone past the Ypt7/HOPS-dependent tethering step.

*4) The effect of lipid composition has been studied previously with a simple lipid mixing system and neuronal SNAREs (Domanska, Kiessling, Tamm, Biophys J. 99, 2936, 2010). However, these experiments only considered the effect on docking and lipid mixing at a fixed SNARE concentration. Interestingly, in these experiments, PE had a reducing role of docking/lipid mixing. Clearly, the use of a lipid-mixing assay may have masked the effects on true fusion, so that could be one explanation of the difference. A discussion of the Domanska reference would be appropriate*.

This has now been added in the Discussion.

*5) The results apply to the vacuolar fusion system – for neuronal fusion, the observed SNARE-protein concentration in synaptic vesicles and in active zones is much higher. Nevertheless, the implications of a critical role of PE, DAG, ERG could also apply to the neuronal system since in that case much faster fusion (msec as opposed to min) is required and it is precisely Ca2+ regulated. Some speculations along these lines might be useful*.

This possibility has now been added in the Discussion.

*6) Although the lipid compositions of vacuoles may have already been well documented elsewhere, it would be helpful to the reader to specifically describe the lipid content of yeast vacuoles so that it can be compared with the standard lipid mixture used in the reconstitutions since the composition of the proteoliposomes is the central theme of this paper. If this standard lipid mixture is an accurate reflection of that found in vacuoles, it would be informative to explicitly make that statement*.

This is now explicitly stated in the 1st paragraph of the Results.

*7) The first sentence of the Discussion suggests the possibility that trans SNARE complex formation induces fusion inexorably. While this view is widely held by many people in the field, I think it would worth mentioning alternate views, in particular because work in Axel Brunger's lab has provided very strong evidence for formation of (partial) trans neuronal SNARE complexes between vesicles without fusion (vesicles can be docked through the SNAREs for 30 minutes without complete fusion!)*.

Done, as suggested.

*8) In the Results section, the text describes a protein-to-lipid ratio of 1:2000, citing*
Table 1*, but*
Table 1
*describes 1:1000. I realize that this apparent discrepancy comes from the fact that only about half of the SNAREs are oriented toward the outside of the vesicles, but readers may still be confused about what the 1:2000 ratio described in the text really means*.

We've now clarified this in the text.